# In Situ Synthesis of Cu_2_O Nanoparticles Using *Eucalyptus globulus* Extract to Remove a Dye via Advanced Oxidation

**DOI:** 10.3390/nano14131087

**Published:** 2024-06-25

**Authors:** Pablo Salgado, Olga Rubilar, Claudio Salazar, Katherine Márquez, Gladys Vidal

**Affiliations:** 1Departamento de Ingeniería Civil, Facultad de Ingeniería, Universidad Católica de la Santísima Concepción, Concepción 4090541, Chile; psalgado@ucsc.cl; 2Centro de Excelencia en Investigación Biotecnológica Aplicada al Medio Ambiente (CIBAMA-BIOREN), Facultad de Ingeniería y Ciencias, Universidad de La Frontera, Temuco 4811230, Chile; olga.rubilar@ufrontera.cl; 3Departamento de Ingeniería Química, Universidad de La Frontera, Av. Francisco Salazar 01145, Casilla 54-D, Temuco 4811230, Chile; 4Centro de Investigación de Polímeros Avanzados (CIPA), Concepción 4051381, Chile; c.salazar@cipachile.cl; 5Centro de Estudios en Alimentos Procesados (CEAP), Campus Lircay, Talca 3460000, Chile; kmarquez@ceap.cl; 6Grupo de Ingeniería y Biotecnología Ambiental (GIBA-UDEC), Facultad de Ciencias Ambientales, Universidad de Concepción, Concepción 4070386, Chile; 7Water Research Center for Agriculture and Mining (CRHIAM), ANID Fondap Center, Victoria 1295, Concepción 4070411, Chile

**Keywords:** LED-assisted photocatalysis, Fenton-like heterogeneous, methylene blue, phenolic compounds, cellulose-based fabric

## Abstract

Water pollution, particularly from organic contaminants like dyes, is a pressing issue, prompting exploration into advanced oxidation processes (AOPs) as potential solutions. This study focuses on synthesizing Cu_2_O on cellulose-based fabric using Eucalyptus globulus leaf extracts. The resulting catalysts effectively degraded methylene blue through photocatalysis under LED visible light and heterogeneous Fenton-like reactions with H_2_O_2_, demonstrating reusability. Mechanistic insights were gained through analyses of the extracts before and after Cu_2_O synthesis, revealing the role of phenolic compounds and reducing sugars in nanoparticle formation. Cu_2_O nanoparticles on cellulose-based fabric were characterized in terms of their morphology, structure, and bandgap via SEM-EDS, XRD, Raman, FTIR, UV–Vis DRS, and TGA. The degradation of methylene blue was pH-dependent; photocatalysis was more efficient at neutral pH due to hydroxyl and superoxide radical production, while Fenton-like reactions showed greater efficiency at acidic pH, primarily generating hydroxyl radicals. Cu_2_O used in Fenton-like reactions exhibited lower reusability compared to photocatalysis, suggesting deterioration. This research not only advances understanding of catalytic processes but also holds promise for sustainable water treatment solutions, contributing to environmental protection and resource conservation.

## 1. Introduction

With the rapid progress of society, the escalating prevalence of pesticides, dyes, antibiotics, and other organic pollutants in the environment has become increasingly alarming, posing significant risks to both human health and aquatic ecosystems [1]. Dyes find extensive application in diverse industries, including textiles, food, plastics, printing, leather, cosmetics, and pharmaceuticals, leading to the generation of substantial volumes of wastewater containing these colorants. The presence of such dyes in water and wastewater presents a formidable threat to human health and the environment given that most of them exhibit toxicity and mutagenicity [2]. Consequently, the development of effective and environmentally sustainable technology for treating water contaminated with dyes is imperative.

The treatment of wastewater typically involves a combination of various mechanical, biological, physical, and chemical processes. Depending on the specific needs of the wastewater in question, processes such as filtration, flocculation, sterilization, or chemical oxidation of organic pollutants are integrated in a customized manner to achieve the desired treatment outcomes [3]. Recent studies have emphasized advanced oxidation processes (AOPs) [4]. These methods facilitate the in situ generation of reactive oxygen species (ROS), including hydroxyl radicals (·OH) and superoxide radicals (·O_2_^−^), effectively enhancing the degradation efficiency. This enhancement holds the potential to achieve complete conversion of the targeted pollutant into harmless byproducts such as CO_2_, H_2_O, and mineral acids [4]. Within the realm of AOPs, there is a notable focus on heterogeneous visible-light photocatalysis and certain Fenton processes that can function under mild conditions. The quest for effective catalysts is crucial for the practical application of these processes and has garnered significant attention [5].

The Fenton process utilizes Fe^2+^ as the catalyst and hydrogen peroxide (H_2_O_2_) as the oxidant. However, a drawback of this process is the substantial formation of iron sludge at pH levels ≥ 4 due to the precipitation of ferric hydroxide [2,6]. Consequently, the Fenton process is confined to a narrow pH range to mitigate the production of solid wastes [6]. In addressing the limitations of the Fenton process, extensive research has been directed towards alternative catalysts [2]. Copper, with its multiple valences (Cu^0^, Cu^+^, and Cu^2+^), emerges as a promising catalyst for Fenton-like reactions owing to its favorable redox ability, higher solubility of Cu^+^ and Cu^2+^ in water compared to Fe^3+^, and reduced generation of metal solid wastes [2]. The Fenton-like redox chemistry involving copper catalysis has been extensively examined. In this mechanism, the Cu(I)/Cu(II) half-cycle facilitates the reduction of H_2_O_2_ to ·OH and OH^−^ (Equation (1)). Simultaneously, the Cu(II)/Cu(I) half-cycle enables the oxidation of H_2_O_2_ to hydrogen peroxide radicals (HO_2_·) or ·O_2_^−^ depending on the pH of the system, constituting the speed-limiting step (Equation (2)) [7]. Notably, the kinetics difference observed in the copper Fenton-like process is approximately 50–100 times less than that of the ferrous Fenton process, rendering copper Fenton-like catalysis significantly more efficient [7].
(1)$${\mathrm{Cu}}^{+}+H_{2}O_{2}\to {\mathrm{Cu}}^{2+}+\cdot \mathrm{OH}+{\mathrm{OH}}^{-}$$
(2)$${\mathrm{Cu}}^{2+}+H_{2}O_{2}\to {\mathrm{Cu}}^{+}+\cdot O_{2^{-}}+{2H}^{+}$$

Semiconductor photocatalysis, a prominent AOP, has gained popularity for decomposing organic pollutants into harmless compounds by generating ROS. Lopis et al. [8] describe the photocatalysis as a process that involves the illumination of semiconductor particles in water, leading to the creation of electron–hole pairs (e^−^/h^+^). These pairs induce oxidation and reduction reactions at the valence band (VB) and conduction band (CB), respectively, resulting in ROS generation. This technology holds promise for its ability to completely degrade pollutants without causing secondary pollution. However, the effectiveness of the process relies on the simultaneous reactivity of both h^+^ and e^−^ with their respective reactants, determined by the oxidation/reduction potentials at the valence band (VB) and conduction band (CB) of the photocatalyst. If VB or CB lacks sufficient oxidation/reduction potentials, the photocatalyst remains inactive.

Metal oxide semiconductors, such as TiO_2_ and ZnO, have been studied as photocatalysts [9]. However, the conventional method of achieving photocatalytic degradation of pollutants involves the use of ultraviolet lamps, primarily mercury vapor lamps. These lamps have drawbacks such as high energy consumption, hazardous mercury content, the need for cooling, short lifespan, and operational challenges [10]. Solar light, while an alternative, presents challenges such as the requirement of a large area, high installation costs, and limitation to daylight hours, hindering the development of photocatalytic reactors and processes [10]. To address the need for energy-efficient sources, the emergence of solid-state technology has led to the development of compact, cost-effective, and environmentally friendly light-emitting diodes (LEDs). LEDs enhance the flexibility of reactor design, eliminating the constraints associated with the shape of mercury lamps, and offer versatility in emitting light of different wavelengths (visible or near-ultraviolet) based on the composition and condition of the semiconducting materials [3]; hence, identifying a visible-light-driven photocatalyst with high efficiency is a critical priority.

Cuprous oxide (Cu_2_O), characterized by a direct bandgap ranging from 2.0 to 2.2 eV (visible light), is a p-type semiconductor with promising potential for applications in visible light-driven processes. These applications include solar energy conversion, hydrogen production, gas sensors, and the degradation of organic contaminants [11]. Utilizing Cu_2_O as a photocatalyst offers various advantages. First, its intrinsic characteristics, including low toxicity, environmental friendliness, and affordability, position it as a promising candidate for photocatalytic applications [9]. Second, the band gap of Cu_2_O can be adjusted for direct utilization of visible light [11]. Third, Cu_2_O demonstrates effective molecular oxygen adsorption, facilitating the scavenging of photo-generated electrons. This process inhibits the recombination of electron–hole pairs, thereby enhancing the overall photocatalytic efficiency [9]. These characteristics make Cu_2_O a promising green technology for wastewater treatment.

For Cu_2_O synthesis, various methods, including electrodeposition, precipitation, hydrothermal synthesis, liquid phase reduction, solvothermal synthesis, thermal oxidation, and microwave-assisted heating, are employed [12]. However, traditional methods involve chemical reducing agents, such as hydrazine hydrate and sodium borohydride, which are costly and require special disposal procedures [13]. In response to these limitations, the green synthesis method of nanoparticles has gained prominence, offering advantages like waste prevention, pollutant removal, use of suitable solvents, and sustainable raw materials [14]. Plant and fruit extracts, rich in compounds like sugars, phenolic compounds, and proteins are harnessed as environmentally friendly reagents for reducing inorganic salts, presenting an alternative to conventional physical and chemical methods for nanoparticle synthesis. Eucalyptus extracts are extensively researched for nanoparticle synthesis due to their global abundance and a high concentration of phenolic compounds [14]. In these environmentally friendly synthesis processes, it has been identified that a high concentration of phenolic compounds in extracts favors the formation and stabilization of nanoparticles [14]. In this same sense, although there is abundant evidence of the use of plant extracts for the synthesis of Cu_2_O [15,16,17], there does not seem to be evidence through which to understand the role of the phytochemicals involved in the formation and stabilization of Cu_2_O. It was also not possible to find research works that address the use of Eucalyptus globulus extracts for the formation of Cu_2_O nanoparticles.

Designing the reactor is a crucial aspect, and researchers are actively exploring two approaches: suspended particle (slurry) reactors and fixed-catalyst reactors. Although rapidly mixed slurry systems excel in contaminant-to-surface mass transfer, concerns arise about potential nanoparticle loss to the treated effluent, raising environmental issues. Nanoparticle separation through membrane filtration adds complexity and cost to the process. While gravity and magnetic separation strategies are under exploration, their practical feasibility remains uncertain. The attention paid to immobilizing a photocatalyst on a suitable substrate has grown because it allows for easy recovery, reusability from water without compromising efficiency, minimizing washout, and preventing aggregation [18]. Utilizing a resilient cellulose matrix is essential for preventing nanoparticle aggregation thanks to its intricate three-dimensional porous structure. Cellulose, the most abundant natural polymer, boasts commendable attributes like biodegradability and biocompatibility. Moreover, the cellulose’s seamless compatibility with metal nanoparticles ensures minimal risk of secondary pollution [19]. However, this approach is constrained by mass transfer limitations, necessitating a maximized surface area for both nanoparticles and support material. Moreover, it introduces geometric challenges in light delivery, prompting the exploration of unconventional reactor designs.

There is a wealth of evidence supporting the use of Cu_2_O nanoparticles for the effective degradation of dye pollutants through photoenergy-driven photocatalysis [9,11,13,15,17,19,20,21] or chemical catalysis, specifically in heterogeneous Fenton-like reactions [2,20,22]. The scientific community generally agrees that immobilized nanomaterials are advantageous in these processes [2,9,18,19,20,23]. However, there has not been exploration of these activities using Cu_2_O nanoparticles synthesized from *E. globulus* plant extracts and immobilized on cellulose supports. This lack of experimentation has resulted in a gap in knowledge regarding the impact of pH on these systems, the effectiveness of visible radiation from LEDs in photocatalytic processes, and the potential for novel reactor designs. Therefore, immobilizing Cu_2_O nanoparticles on cellulose-based supports has the potential to expand their applications beyond enhancing photocatalytic activities to also improving Fenton-like reactions for the efficient degradation of organic pollutants, driven by both photoenergy and chemical potential. The objectives of this study were to develop Cu_2_O nanoparticles using *E. globulus* extracts and immobilize them on cellulose-based supports, such as fabric. The study also aimed to investigate the role of the main phytochemicals in the extracts in the synthesis of Cu_2_O and to characterize the nanoparticles. The immobilized Cu_2_O nanoparticles were then tested for their photocatalytic activity under visible radiation from an LED source, as well for as their catalytic activity in the heterogeneous Fenton-like reaction at both acidic and neutral pH levels. The study also examined the reusability capacity of these systems and the type of ROS they generated.

## 2. Materials and Methods

### 2.1. Preparation of E. globulus Extracts and Synthesis of Cu_2_O on Supports

In the summer, leaves of *E. globulus* were collected from Concepcion in southern Chile. The leaves were then washed with deionized water and dried at 50 °C for 24 h before being crushed. The crushed leaves were heated to 60 °C and refluxed for 20 min at a concentration of 60 g/L. After cooling, the extract was centrifuged at 7000 rpm for 20 min. The resulting supernatant was then vacuum filtered using 0.45 μm pore size filter paper and stored at 2 °C for future use. To synthesize Cu_2_O nanoparticles using *E. globulus* extracts supported on cellulose-based support, the procedure was as follows: A 4 × 4 cm^2^ piece of fabric was immersed in 50 mL of 0.5 mol/L CuCl_2_ solution and stirred at 200 rpm at 60 °C for 30 min. The fabric was then removed from the solution. Next, 10 mL of 1 g/50 mL NaOH solution was added to the CuCl_2_ solution via constant dripping and stirred at 500 rpm to form Cu(OH)_2_. The fabric was immersed in this solution and stirred at 200 rpm for 30 min at 60 °C. After 30 min, the fabric was removed and placed in 25 mL of *E. globulus* extracts at 60 °C and 200 rpm for an additional 30 min. Finally, the fabric was rinsed with deionized water and dried in an oven at 60 °C. Hereafter, the Cu_2_O nanoparticles synthesized on fabric are named F-CuNP.

The *E. globulus* extracts were analyzed using FTIR spectroscopy, employing the Spectrum Frontier/Spotlight 400 Microscopy System (Perkin Elmer, Inc., Waltham, MA, USA) with a linear array of mercury–cadmium–tellurium (MCT) detectors. Spectra were collected in transfer mode, with each extract solution (before and after nanoparticle synthesis) represented by a 0.5 μL drop on a MirrIR slide. After drying for 30 min at 70 °C in an oven, spectra were acquired using the following parameters: an aperture of 100 × 100 μm^2^, a spectral range of 4000–500 cm^−1^, a spectral resolution of 4 cm^−1^, and 64 scans per spectrum. Background spectra were obtained from a low-emissivity area of the glass slide without the materials under analysis and were automatically subtracted from the material spectra. For the analysis of F-CuNP in ATR mode, a germanium crystal (refraction index 4.01) with a pixel size of 6.25 μm × 6.25 μm was used, with a spectral range from 4000 to 750 cm^−1^, a spectral interval of 4 cm^−1^, a spectral resolution of 8 cm^−1^, 64 scans per spectrum, and an aperture of 400 × 400 μm^2^. The spectra underwent atmospheric noise removal and baseline correction.

### 2.2. SEM Analysis

The nanoparticle distribution across the substrate was investigated through the acquisition of SEM images (SU3500, Hitachi, Tokyo, Japan). An X-ray energy-dispersive spectroscopy (EDX)-based detector (QUANTAX 100, Bruker, Ettlingen, Germany) was utilized to observe the elemental distribution on the surface, operating at an acceleration voltage of 30.0 kV and a working distance of 10.8 mm. The calculation of particle size for nanoparticles and the creation of a histogram plot were performed using ImageJ software version 1.53t.

### 2.3. XRD Analysis

The X-ray diffraction (XRD) patterns of the samples were collected using a Bruker D4 ENDEAVOR X-ray diffractometer, with Cu Kα radiation as the source. The operating voltage and current were set at 40 kV and 20 mA, respectively. Diffraction peak intensities were recorded within a 10–80° (2θ) range, with increments of 0.02° and a count duration of 0.3 s per increment. To estimate the crystallite sizes of nanoparticles, the Debye–Scherrer formula (Equation (3)) was used, where D represents the mean crystallite size, K is a dimensionless shape factor (assumed to be 0.94 in this case), λ is the X-ray radiation wavelength (Cu Kα = 0.154056 nm), β denotes the band broadening at half-maximum intensity, and θ is the diffraction angle.
(3)$$D=\frac{K\lambda }{\beta \cos\theta }$$

### 2.4. TGA Analysis

Thermogravimetric analysis (TGA) was performed on sample fragments using a Perkin-Elmer instrument STA 6000. The samples were heated at a rate of 15 °C/min under a nitrogen flow of 40 mL/min, spanning a temperature range of 25–600 °C.

### 2.5. Optical Analysis

The optical characteristics of the supported nanoparticles were evaluated using diffuse reflectance UV–vis spectra (DRS) on a Jasco V-750 UV-Visible spectrophotometer, which is outfitted with Jasco ISV-922 integrating sphere (Tokyo, Japan). The band gap of the copper oxide nanoparticles was ascertained from a Tauc plot derived from the UV–Vis diffuse reflectance spectra.

### 2.6. Phytochemical Analysis before and after Synthesis

The analysis of the *E. globulus* extracts, both before and after nanoparticles formation, included the quantification of phenolic compounds, reducing sugars, and proteins [24]. The Folin–Ciocalteu method was employed to measure the phenolic compounds, with the results expressed as gallic acid equivalents based on a calibration curve of standard gallic acid solutions. The DNS method was used to determine the concentration of reducing sugars, with the values calculated as glucose equivalents using a calibration curve of standard glucose solutions. The Bradford method was utilized to quantify the protein content of the extracts, using bovine albumin serum solutions as the standard. The ferric-reducing antioxidant power method (FRAP) and cupric reducing antioxidant capacity (CUPRAC) [25] were modified and applied to ascertain which phytochemicals in the *E. globulus* extract function as reducing agents in the formation of supported nanoparticles.

### 2.7. MB Degradation by Photocatalysis and Fenton-like Reaction

The experiment on photocatalytic and Fenton-like degradation of methylene blue (MB) at a concentration of 50 mg/L was conducted in a homemade reactor consisting of four borosilicate glass chambers, each with a volume of 20 mL, at room temperature (Figure 1).

The 4 × 4 cm^2^ pieces of F-CuNP were placed onto 3D-printed supports. The solution was stirred continuously in the dark for 120 min to achieve adsorption and desorption equilibrium. After 120 min, the LED light strips were turned on for photocatalysis. For the Fenton-like experiments, 50 mM H_2_O_2_ was added after 120 min in darkness. All the Fenton experiments were conducted in the dark. Every 20 min, 0.5 mL of the solution was sampled, and the absorbance was measured at 664 nm using a UV-vis spectrophotometer (Jasco, V-750). The impact of pH (3.0 and 7.0) on the photocatalytic and Fenton-like activity of F-CuNP was examined. The pH of the aqueous solution was regulated using HCl and NaOH.

The degradation percentage of the dye was calculated using Equation (4):(4)$$\mathrm{MB}\;\mathrm{degradation}\;\left(\%\right)=\frac{{(C}_{0}-{\;C}_{t})}{C_{0}}\times 100,$$
where C_0_ and C_t_ are the initial and final concentrations of MB, t is time (min), and k_app_ (min^−1^) is the first-order apparent rate constant for MB degradation.

After a selection of the best performances obtained in previous studies, the recyclability and stability of F-CuNP samples were assessed after five cycles of degradation using photocatalysis and Fenton-like processes. After each cycle, the samples were rinsed with deionized water and dried in an oven at 50 °C before being used for another degradation reaction. To better understand the degradation mechanism of MB by F-CuNP, active species trapping experiments were conducted using scavengers such as CrO_3_, Na_2_C_2_O_4_, p-benzoquinone (BQ), and isopropanol (IP) as the representative scavengers of e^−^, h^+^, ·O_2_^−^, and ·OH, respectively.

## 3. Results

### 3.1. Synthesis of Nanoparticles Supported

To examine the appearance of cotton fabric and as support before and after the synthesis of Cu_2_O, their digital photos were obtained (Figure 2). The evident change of color indicated on the surface of supports suggest the formation of nanoparticles.

### 3.2. Characterization of Nanoparticles

#### 3.2.1. SEM-EDX Analysis

From SEM images (Figure 3), it can be seen that synthesized F-CuNP are spherical with uniform distribution on the support surface. The size of F-CuNP varied from 34.0 to 173.5 nm, with a mean diameter of 81.84 nm. EDX analyses indicate a concentration of 45.72% C, 50.55% O, and 3.73% Cu for F-CuNP.

#### 3.2.2. XRD Analysis

Figure 4a shows that XRD analysis of F-CuNP reveals the diffraction peaks at 2θ values 29.51°, 36.38°, 42.34°, and 61.39°, corresponding to the lattice planes (110), (111), (200), and (220), respectively. These values agree with the monoclinic Cu_2_O phase (JCPDS N° 05-0667 database) [26]. Through the Debye–Scherrer equation, the average crystalline size of F-CuNP from the four diffraction peaks was determined to be 97.21 nm. The other diffraction peaks close to 25° and 34° are typical of cellulose [27,28].

#### 3.2.3. Raman Analysis

Raman spectra of F-CuNP as a function of wavenumbers ranging from 50 to 1160 cm^−1^ are illustrated in Figure 4b.

The F-CuNP exhibited distinct peaks in the spectra at 113, 135, 162, 218, 271, 308, 392, 445, 505, 548, 624, 787, 818, and 1111 cm^−1^. These peaks confirm the successful preparation of Cu_2_O nanoparticles [29,30,31,32,33,34,35]. These analyses are consistent with the findings from the XRD analyses. Other peaks in the spectra at 180, 324, 483, 522, 580, 694, 745, 767, 850, 886, 911, 1020, 1031 1066, and 1136 cm^−1^ for F-CuNP confirm the presence of various phenolic compounds [36,37]. Signals related to cellulose cotton are exhibited at 343, 378, 427, 971, 996, 1051, 1089, and 1102 cm^−1^ for F-CuNP [38,39].

#### 3.2.4. FTIR Analysis

The FTIR spectrum of pristine fabric and F-CuNP is shown in Figure 5.

The main bands observed in the FTIR analyses of the pristine fabric, as well as in F-CuNP, are summarized in Table 1.

In the FTIR analysis of pristine fabric, signals identified at approximately 3339, 3298, 2899, 2852, 1638, 1458, 1430, 1368, 1338, 1317, 1205, 1162, 1053, 1003, and 900 cm^−1^, mainly attributed to cellulose in the fabric, exhibit a decrease in intensity. This phenomenon has been associated by Mussino et al. [51] with the interaction of cellulose groups with other nanoparticles and by Sedighi et al. with the interaction of cellulose with Cu_2_O-NP [27]. Additionally, it has been suggested that this decrease is linked to the interaction between the functional groups of cellulose and phytochemicals present in the *E. globulus* extract [52].

Conversely, in the FTIR analyses of F-CuNP, certain signals emerge that were not present in the pristine fabric samples. Bands indicating this behavior are observed at 1506, 1409, 1176, 850, and 793 cm^−1^, characteristic of phenolic compounds, a band at 1117 cm^−1^, characteristic of proteins, a band at 1472 cm^−1^, characteristic of lipids, and a band at 874 cm^−1^, attributed to the presence of glucose, possibly due to the interaction of these phytochemicals with the Cu_2_O-NPs [53]. Another band close to 1717 cm^−1^ is attributed to the presence of C=O bonds, possibly arising from the oxidation of -OH groups to C=O during the reduction of Cu^2+^ ions to Cu^+^ for the subsequent formation of Cu_2_O [54].

Furthermore, characteristic cellulose signals around 1108, 1026, and 986 cm^−1^ are discernible, showing a slight shift when comparing the spectra of pristine supports with those incorporating Cu_2_O-NPs. This shift could signify the interaction of these groups either with the Cu_2_O-NPs or with the phytochemicals from the *E. globulus* extract [27,28].

#### 3.2.5. TGA Analysis

The TGA was carried out to study the thermal behavior of pristine fabric and F-CuNP samples. To enhance comprehension of the results, derivative thermograms are graphically represented in conjunction with the thermograms. The results are shown in Figure 6.

The mass of samples decreased as the temperature increased from 250 to 600 °C, consistent with the chemical nature of cellulose [55]. The derivative thermograms distinctly reveal multiple degradation events in the samples. The first stage occurs at a temperature below 100 °C, due to the loss of water, which could be strongly bound to the cellulosic fibers due to its hydrophilic character [56]. Notably, the primary degradation of the cellulose matrix is evident between 350 and 370 °C, attributed to the decarboxylation, decomposition, and depolymerization of glycosyl units within cellulose during heating scans [56,57]. Low-intensity peaks were observed at 417.6 °C and 423.1 °C for samples with fabric, which can be attributed to the release of volatiles because of the decomposition of by-products at previous stage [56,57]. Intriguingly, the degradation temperature of cellulose decreases in the F-CuNP sample. This reduction could be attributed to the catalytic activity of Cu_2_O-NP, accelerating the thermal degradation of cellulose [48]. Furthermore, the appearance of a mass loss stage at 315.6 °C for the F-CuNP sample has been attributed to the phase transition from Cu_2_O to CuO [58]. In this same sense, Balık et al. [35] demonstrated through XRD, FTIR, and Raman analysis that Cu_2_O nanoparticles begin a phase transition to CuO from 250 to 350 °C.

#### 3.2.6. Optical Analysis

The direct optical bandgap values of F-CuNP were determined via Tauc plot [59]. The bandgap values (E_g_) of F-CuNP are calculated as 2.64 eV (Figure 7a).

The conduction and valence band positions of Cu_2_O on F-CuNP can be determined using the following empirical relations (Equations (5) and (6)) [60]:(5)$$E_{\mathrm{CB}}=\chi -E^{e}-0.5E_{g},$$
(6)$$E_{\mathrm{VB}}=E_{\mathrm{CB}}+E_{g},$$
where ECB is the conduction band potential; EVB is the valence band potential; E_g_ is the bandgap of the semiconductor; χ is the absolute electronegativity of the semiconductor, which is defined as the arithmetic mean of the atomic electron affinity and the first ionization energy [61] (5.32 eV for Cu_2_O [62]); and E^e^ is the energy of free electrons on the hydrogen scale (~4.5 eV) [60]. Therefore, the calculated values of ECB and EVB for F-CuNP are −0.50 and 2.14 eV, respectively (Figure 7b).

To maximize the utilization of electrons and holes in redox reactions, the conduction band (CB) of Cu_2_O should be significantly higher than the reduction potential of O_2_/·O_2_^−^ and H_2_O_2_/O_2_, and the valence band (VB) should be substantially lower than the oxidation potential of OH^−^/·OH [63]. The results in Figure 7b indicate that the e^−^ in the CB of the synthesized Cu_2_O nanoparticles could be trapped by O_2_ in the medium to produce ·O_2_^−^ and H_2_O_2_, while the h^+^ in the VB could react with OH^−^ to produce ·OH.

Cu_2_O is a p-type semiconductor with direct bandgap energy of about 2.0–2.2 eV [19,23,64], which indicates an unusual increase in the bandgap energy detected in the Cu_2_O samples synthesized on F-CuNP. It has been proposed that the shift in the bandgap energy of semiconductors could be related mainly to changes in the size of the particles [64,65,66], the effect of the exposed facets in the crystalline structure of the particles [65,67], the effect of ligands that interact with particles [68,69], and defects in the materials [70,71,72].

As already mentioned, there is abundant evidence regarding the effect of particle size and the exposed facets in the crystalline structure on the bandgap of semiconductors. However, Huang et al. [65] describe the effect of both factors together on the optical properties of Cu_2_O nanoparticles. The authors confirm, after studying different sizes and structures of Cu_2_O nanoparticles, that at smaller Cu_2_O sizes, their bandgap increases for each of the exposed facets in the crystalline structure. Furthermore, it is notable that the authors found that Cu_2_O nanoparticles with the exposed facet (111) and diameters of 85 and 117 nm exhibit a bandgap of 2.70 and 2.61 eV, respectively, similar to what was found in this investigation. Accordingly, these findings could indicate a strong influence of the particle size and the exposed facet on the bandgap of synthesized F-CuNP.

### 3.3. Identification of Biomolecules Involved in the Synthesis of Cu_2_O Nanoparticles

#### 3.3.1. Spectrophotometric Analysis of Extracts before and after Formation Cu_2_O Nanoparticles

Figure 8 reveals the changes in the main phytochemicals contained in the *E. globulus* extract before and after the formation of F-CuNP.

First, it is notable that no significant changes occur in the concentration of reducing sugars or in the concentration of proteins when comparing the initial and post-synthesis of Cu_2_O nanoparticles. Several authors have carried out similar studies, concluding that these proteins do not seem to actively participate in the reduction of Cu^2+^ to form Cu_2_O [73] or other nanoparticles [74,75]. Regarding the role of sugars in leaf extracts for the Cu_2_O nanoparticle synthesis process, there are some authors who attribute an ion-reducing role to sugars [16,17], while others attribute more of a stabilizing role to these phytochemicals [73,76]. Otherwise, the changes in the concentrations of phenolic compounds before and after the formation of Cu_2_O nanoparticles were significant. There is abundant evidence regarding the role that phenolic compounds play in the formation of Cu_2_O nanoparticles, acting as reducers or stabilizers [45,73,76,77]. It is possible to observe a similar behavior when the results of FRAP and CUPRAC are analyzed since both methods used in the characterization of plant extracts attempt to quantify the antioxidant capacity based mainly on the presence of phenolic compounds [25]. Consequently, this result would again suggest the main role of phenolic compounds in the formation of Cu_2_O nanoparticles.

#### 3.3.2. FTIR Analysis of Extracts before and after Formation Cu_2_O Nanoparticles

A comparative experiment was conducted to examine the formation of Cu_2_O nanoparticles on fabric substrates using *E. globulus* extract. The objective of this experiment was to understand the role of different components of plant extracts in the synthesis process. FTIR measurements were performed to identify the primary functional groups involved in the formation of Cu_2_O. Figure 9 shows the FTIR spectra of the *E. globulus* extract before and after the synthesis of Cu_2_O on the fabric substrates. A considerable decrease is observed in the signals ~3375 cm^−1^, indicating the presence of alcoholic and phenolic -OH groups [50]; ~2936 cm^−1^, attributed to C–H asymmetric stretching [50]; ~1700 cm^−1^, attributed to C=O stretching; the band ~1610 [50], correlated with C=C stretching; the signal at ~1350 cm^−1^, attributed to C–H bending modes [54]; ~1450 cm^−1^, associated with C=C aromatic ring stretching [45]; ~1400 cm^−1^, attributed to C-OH stretching vibrations [45]; ~1250 cm^−1^, attributed to C–O stretching (in aromatic esters) [50]; and 1060 cm^−1^, attributed to C–O–C stretching asymmetric vibration [76]. All these signals are clearly attributed to different phenolic compounds in *E. globulus* extracts, and after the synthesis of nanoparticles, these phytochemicals decrease in concentration.

### 3.4. Photocatalytic and Fenton-like Degradation of MB

A series of experiments were carried out to compare the performance of F-CuNP in the degradation of MB at pH 3.0 and 7.0 via photocatalysis and a heterogeneous Fenton-like reaction.

Figure 10a shows the photocatalytic degradation kinetics of MB catalyzed by F-CuNP at pH 3.0 and 7.0. It is clearly noticeable that F-CuNP at pH 7.0 exhibits photocatalytic capacity greater than at pH 3.0, with apparent rate constants (k_app_) of 0.0233 and 0.00377 min^−1^, respectively (Figure 10b). These results indicate a clear effect of the pH. Accordingly, the fact that the greatest degradation of MB is observed at basic pH could be related to the interactions between Cu_2_O and MB. Akter et al. [78] found similar results when studying the effect of pH on MB removal using Cu_2_O. These authors have postulated a greater influence of electrostatic interactions, compared to another interaction, between MB and Cu_2_O at basic pH as the reason for greater removal. Specifically, it would be associated with a greater negative charge on the Cu_2_O at higher pH and a greater cationic character of the MB at higher pH. These differences in the charges of the dye and the Cu_2_O nanoparticles would favor the AM to be better adsorbed on the Cu_2_O surface, causing a higher degradation rate. In this sense, making the same comparison but only in the dark adsorption process, it is possible to observe that for F-CuNP, a greater amount of AM is absorbed at pH = 7.0 than at pH = 3.0, which would also agree with what was previously postulated. This pH effect was also found in systems with other adsorbents to remove MB [79,80] and with Cu_2_O nanoparticles to remove other contaminants [81].

Figure 10c shows the degradation of MB using a heterogeneous Fenton-like process using F-CuNP at pH 3.0 and 7.0. Once again, it is possible to observe a clear effect of the pH in the MB removal. It is possible to indicate that in the system using F-CuNP at pH 3.0, the MB removal is greater than at pH 7.0 with k_app_ values of 0.0347 and 0.00722 min^−1^, respectively (Figure 10d). Similar effects with respect to pH have been detected in systems that include the use of Cu_2_O for Fenton-like reactions [2,20,82,83,84,85]. The reasons that have been attributed to the greater degradation of contaminants in Fenton-like systems using Cu_2_O are varied: (i) at acidic pH, the dissolution of Cu_2_O is favored, forming Cu(I) ions that react quickly with H_2_O_2_ to produce ·OH according to Equation (1) [20,82,83,84]; (ii) H_2_O_2_ could decompose at high pH, preventing it from reacting and forming ·OH [20,85]; and (iii) ·OH could react with OH^−^ ions, decreasing the availability of ·OH to degrade MB [86].

### 3.5. Reusability of Photocatalytic and Fenton-like Degradation of MB

The reusability studies were conducted to study the recyclability of F-CuNP in the removal of MB dye solution at pH 7.0 for photocatalysis and at pH 3.0 for Fenton-like reaction (Figure 11). The photocatalytic degradation efficiency for F-CuNP after five cycles of reuse (Figure 11a) decreased from 93.9% to 86.5%, while for the Fenton-like reaction (Figure 11b), the degradation efficiency after five cycles of reuse decreased from 99.36% to 77.38% at pH = 3.0. It is possible to indicate that the reusability efficiency is longer in photocatalytic systems, with a less abrupt decline in the efficiency of AM degradation than for AM degradation by Fenton-like systems. The lower performance of F-CuNP for Fenton-like systems throughout the reuse cycles could be associated with the greater dissolution of Cu_2_O at acidic pH [22] or with the reaction of Cu^+^ on the Cu_2_O surface with H_2_O_2_, causing a lower availability as Cu^+^ are consumed and converted into Cu^2+^ [5].

### 3.6. Reactive Species in Photocatalytic and Fenton-like Degradation of MB

To propose a mechanism for the photocatalytic and Fenton-like degradation of MB, the reactive species in these processes were investigated (Figure 12). Instead of measuring the degradation efficiency in a solution containing MB and F-CuNP, it is possible to detect the first step of photocatalysis via the generation of e^−^ and h^+^ using scavengers. Similarly, it is possible to detect the radicals ·OH and ·O_2_^−^ generated in the photocatalytic system using scavengers. Thus, CrO_3_ was used as an e^−^ scavenger, Na_2_C_2_O_4_ was add as an h^+^ scavenger, p-benzoquinone (BQ) as an ·O_2_^−^ scavenger, and isopropanol (IP) as an ·OH scavenger.

Regarding the effect of scavengers in Fenton-like systems at pH = 3.0 for F-CuNP, the effect of BQ as an ·O_2_^−^ scavenger causes a decrease in the MB degradation efficiency of 16.95% for F-CuNP, respectively, compared to the system without scavengers (Figure 12b). However, the use of IP as an ·OH scavenger induces a more notable decrease in MB degradation efficiency of 71.58% for F-CuNP. This greater decrease in degradation efficiency when using IP indicates a greater influence of ·OH on MB degradation than ·O_2_^−^, similar to what has been found in other works using Cu_2_O/H_2_O_2_ at acidic pH [20,87]. There are several things that can explain this behavior: (i) ·OH (1.5–2.8 V) [88] has a higher redox potential than ·O_2_^−^ (·O_2_^−^/H_2_O_2_ = +0.94 V, O_2_/·O_2_^−^ = −0.35 V) [89]; (ii) the reaction between Cu^+^ and H_2_O_2_ (Equation (1); 1 in Figure 13) has a higher rate constant to produce ·OH and Cu^2+^ (k = 4 × 105 M^−1^ s^−1^ at pH 6–8) than the reaction between Cu^2+^ and H_2_O_2_ that produces ·O_2_^−^ and Cu^+^ (k ≤ 1 M^−1^ s^−1^ at pH 7) (Equation (2); 2 in Figure 13) [90]; and (iii) the interaction of H_2_O_2_ on the surface of Cu_2_O causes the breaking of the O-O bond of H_2_O_2_ to generate ·OH [20]. It is worth mentioning that the possible reaction pathways involving Cu^+^, Cu^2+^, and H_2_O_2_ could occur with the copper ions bound to the surface of the Cu_2_O nanoparticles (=Cu^+^ and =Cu^2+^) as well as with the dissolved ions.

Otherwise, for the effect of scavengers on photocatalytic processes at pH = 7.0 (Figure 12a), using Na_2_C_2_O_4_ slightly affects the degradation efficiency of the MB, decreasing only 11.8% for F-CuNP with respect to the systems without scavengers. On the contrary, the use of CrO_3_ significantly affects the MB degradation efficiency, decreasing by 60.94% for F-CuNP with respect to systems without scavenger. These results suggest an effective migration of the photoexcited e^−^ to the Cu_2_O surface, causing CrO_3_ to be effective in its inhibition. On the other hand, the effect of Na_2_C_2_O_4_ does not seem relevant, which seems to indicate the low availability of h^+^ on the Cu_2_O surface to degrade MB or react with OH^−^ (7 and 8 in Figure 13). These results could indicate that reduction reactions would be more important than oxidation reactions. Regarding the effect of radical scavengers, the presence of BQ causes a decrease of 32.3% for F-CuNP in terms of the degradation efficiency of MB with respect to systems without scavengers. Otherwise, the use of IP showed a significantly greater decrease in MB degradation of 81.08% for F-CuNP in relation to the system evaluated without scavengers. Because the action of IP is important, the production and subsequent action of ·OH in the degradation of MB seems to be more relevant than the action of the ·O_2_^−^ produced. Similarly, Bayat and Sheibani [91], using a CuO/Cu_2_O heterostructure as a photocatalyst under visible radiation to degrade MB, also discovered that the main species responsible for the degradation of this dye are the ·OH and e^−^ produced [91].

Based on what has already been mentioned, it is possible to suggest a low presence of h^+^ on the Cu_2_O surface, so the ·OH, the main reactive species that is responsible for degradation of MB would not be produced by the reaction of the h^+^ and OH^−^ ions (7 and 8 in Figure 13). On the other hand, the high presence of e^−^ on the Cu_2_O surface and the activity of ·O_2_^−^ with respect to the degradation of MB could support the assumption that ·OH radicals are being produced by parallel reactions (Equations (7)–(13), 3, 4, 5, and 6 in Figure 13) and not because of the h^+^ oxidizing OH^−^ [21,78,91].
(7)$${\mathrm{Cu}}_{2}O+h\nu \to {\mathrm{Cu}}_{2}O\;(h^{+}+e^{-})$$
(8)$$O_{2}+e^{-}\to {\cdot O}_{2}^{-}$$
(9)$$O_{2}+{2H}_{2}O+2e^{-}\to H_{2}O_{2}+{2\mathrm{OH}}^{-}$$
(10)$${2H}^{+}+O_{2^{-}}\to H_{2}O_{2}$$
(11)$$H_{2}O_{2}+e^{-}\to \;\cdot \mathrm{OH}+{\mathrm{OH}}^{-}$$
(12)$$H_{2}O_{2}\to 2\cdot \mathrm{OH}$$
(13)$$H_{2}O_{2}+O_{2^{-}}\to \cdot \mathrm{OH}+{\mathrm{OH}}^{-}+O_{2}$$

Chu and Huang [92], in a study on facet-dependent photocatalytic properties of Cu_2_O using scavengers, found results and trends similar to those of the present study for Cu_2_O nanoparticles with a predominantly octahedral structure. The authors suggest that the photocatalytic activity of octahedral Cu_2_O arises mainly from the migration of e^−^ to the catalyst surface to produce ·O_2_^−^ and ·OH and not from the weak migration of h^+^ to the surface.

## 4. Conclusions

Cu_2_O nanoparticles were synthesized in situ on cellulose-based fabric using *E. globulus* extracts, resulting in sizes below 100 nm. Raman and FTIR analyses demonstrate the interaction between phenolic compounds from the extracts and Cu_2_O nanoparticles. Additionally, FTIR shows the formation of C=O bonds, indicating the oxidation of -OH groups in phenolic compounds to produce Cu_2_O. Furthermore, FTIR reveals potential interactions between the cellulose and Cu_2_O or phenolic compounds. TGA studies confirm the presence of Cu_2_O in F-CuNP, as well as the accelerated cellulose degradation in their presence. Optical properties analyses suggest the potential for generating ·OH and ·O_2_^−^ through photogenerated e^−^ and h^+^. The effect of size and facet exposure on the Cu_2_O bandgap is also proposed. Phytochemical analyses imply the involvement of phenolic compounds in the reduction of Cu^2+^ to Cu^+^ and the stabilization of nanoparticles. MB degradation is more significant via photocatalysis at pH = 7.0 and Fenton-like reaction at pH = 3.0 using F-CuNP. The reusability of these systems is high, but it declines faster in Fenton-like systems due to Cu_2_O nanoparticle decomposition at acidic pH. Photocatalytic systems primarily degrade MB through ·OH and e^−^, while Fenton-like systems are influenced by ·OH generation. This research highlights the feasibility of obtaining Cu_2_O nanoparticles immobilized on cost-effective supports, functioning as both photocatalysts and Fenton-like reaction catalysts. These nanoparticles can be activated by visible light and synthesized in an eco-friendly manner.

## Figures and Tables

**Figure 1 nanomaterials-14-01087-f001:**
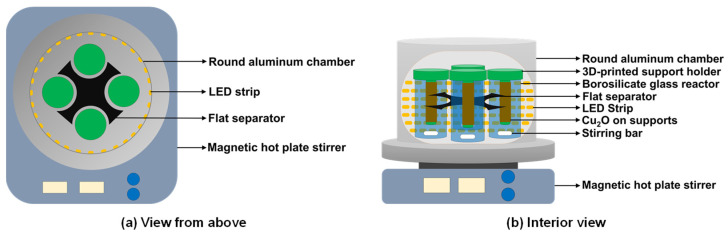
View from above and inside the reactor used for degradation of MB via photocatalysis and Fenton-like reaction.

**Figure 2 nanomaterials-14-01087-f002:**
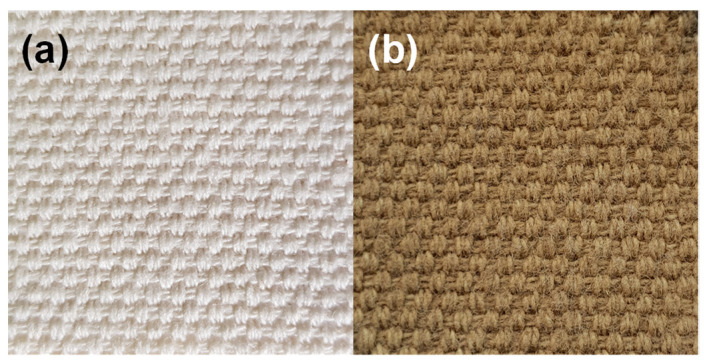
Photographic images of (**a**) pristine fabric and (**b**) F-CuNP.

**Figure 3 nanomaterials-14-01087-f003:**
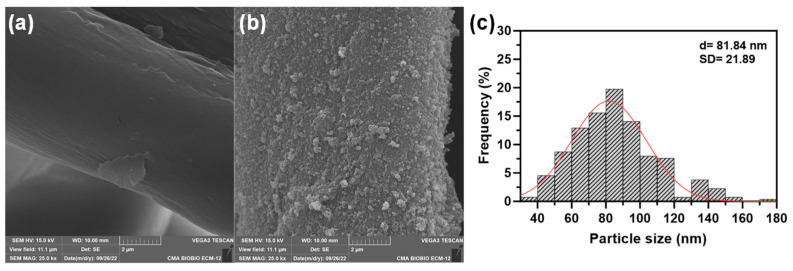
SEM analysis for (**a**) pristine fabric, (**b**) F-CuNP, and (**c**) histogram of particles size of F-CuNP.

**Figure 4 nanomaterials-14-01087-f004:**
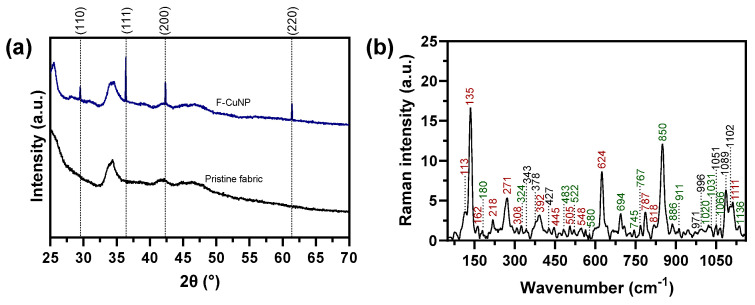
(**a**) XRD patterns of F-CuNP and pristine fabric; (**b**) Raman spectra for F-CuNP (red numbers for Cu_2_O nanoparticles signals, green numbers for phenolic compounds signals, and black numbers for cellulose signals).

**Figure 5 nanomaterials-14-01087-f005:**
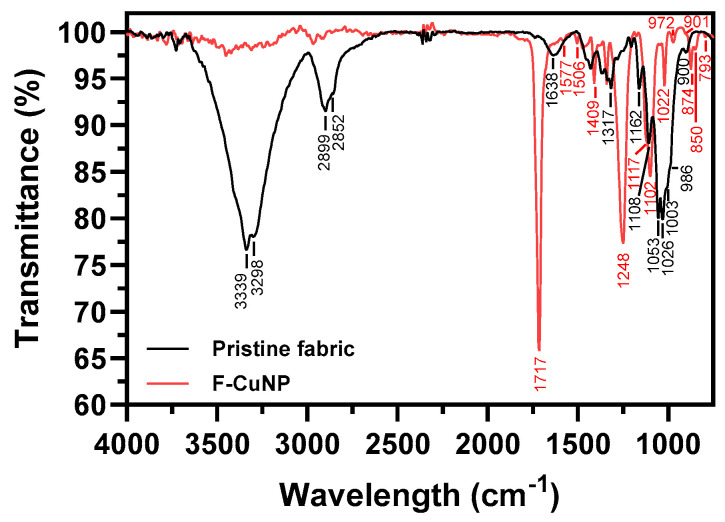
FTIR analyses for pristine fabric and F-CuNP.

**Figure 6 nanomaterials-14-01087-f006:**
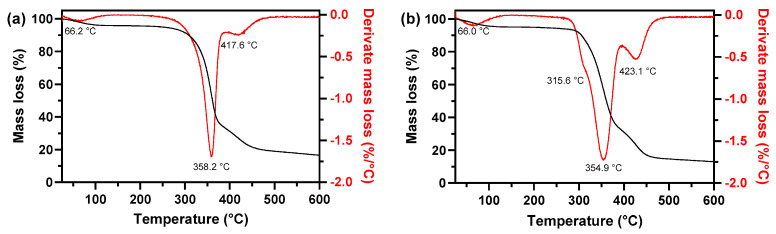
TGA curves of (**a**) pristine fabric and (**b**) F-CuNP.

**Figure 7 nanomaterials-14-01087-f007:**
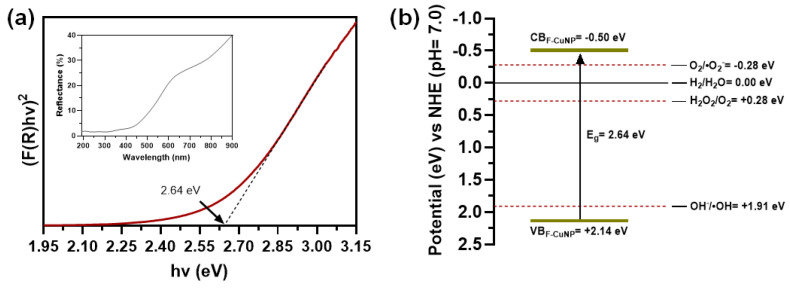
(**a**) Tauc plot and DRS spectra (inset graph) and (**b**) potential band diagram for F-CuNP.

**Figure 8 nanomaterials-14-01087-f008:**
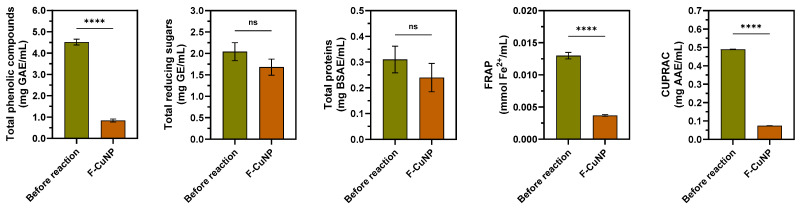
Spectrophotometric analysis of total phenolic compounds, total reducing sugars, total proteins, FRAP, and CUPRAC capacity in the extract of *E. globulus* before and after the synthesis of F-CuNP. The data obtained are presented as mean ± standard deviation. Differences among groups were compared via ANOVA with Tukey post hoc analysis. **** *p* < 0.0001, ns: no significant differences.

**Figure 9 nanomaterials-14-01087-f009:**
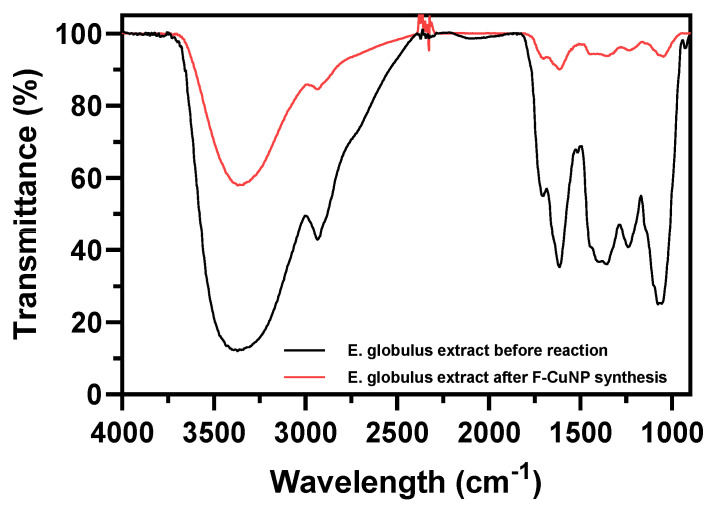
FTIR spectra of *E. globulus* extract before and after synthesis of F-CuNP.

**Figure 10 nanomaterials-14-01087-f010:**
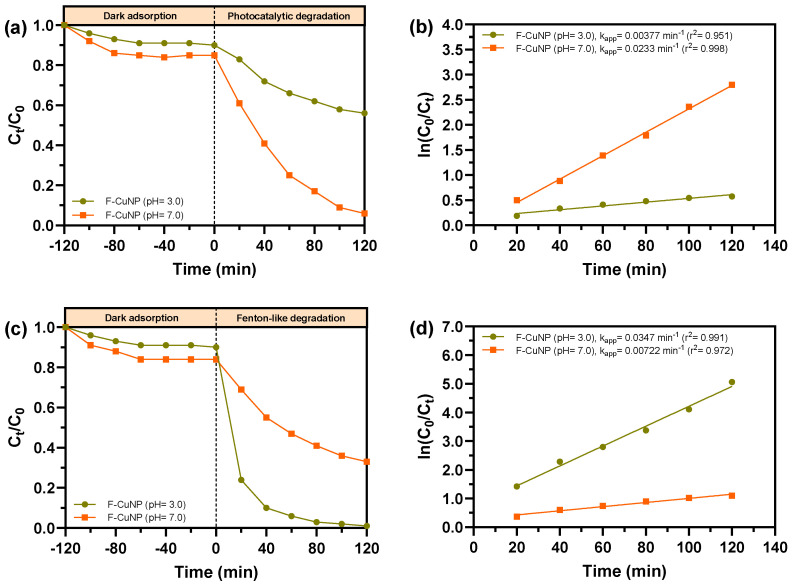
(**a**) Photocatalytic under LED visible light and (**c**) Fenton-like degradation efficiency of MB at different time intervals catalyzed by F-CuNP at pH 3.0 and 7.0 (inset: legend represents samples at different pH). (**b**) Photocatalytic and (**d**) Fenton-like calculated degradation rate constant kapp for F-CuNP at pH 3.0 and 7.0 (inset: legend represents samples at different pH, k_app_ value, and r^2^ of pseudo-first-order model).

**Figure 11 nanomaterials-14-01087-f011:**
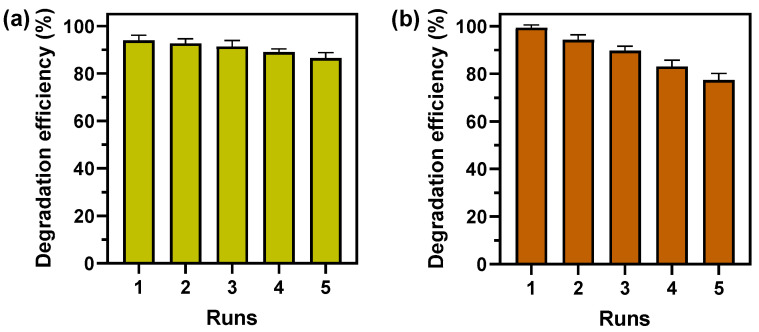
Reusability study of MB degradation using F-CuNP (**a**) via photocatalysis under LED visible light for F-CuNP at pH = 7.0 and (**b**) via Fenton-like reaction for F-CuNP at pH = 3.0 in darkness. The data obtained are presented as mean ± standard deviation.

**Figure 12 nanomaterials-14-01087-f012:**
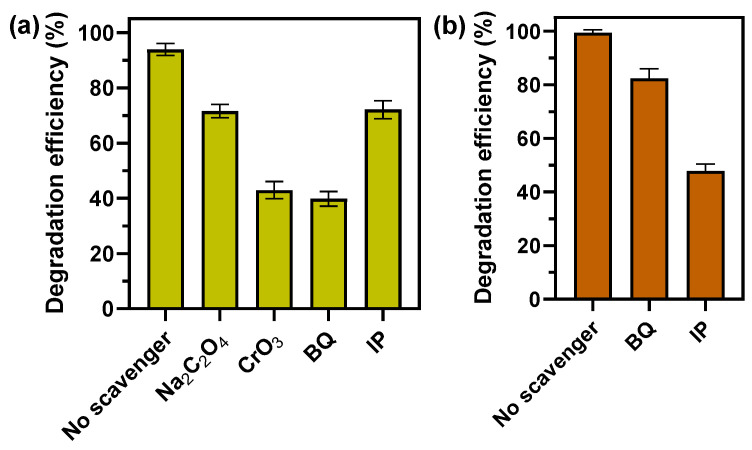
(**a**) Effect of Na_2_C_2_O_4_, CrO_3_, BQ, and IP scavengers on photocatalytic degradation of MB over F-CuNP under LED visible light; (**b**) effect of BQ and IP scavenger on Fenton-like degradation of MB over F-CuNP in darkness. The data obtained are presented as mean ± standard deviation.

**Figure 13 nanomaterials-14-01087-f013:**
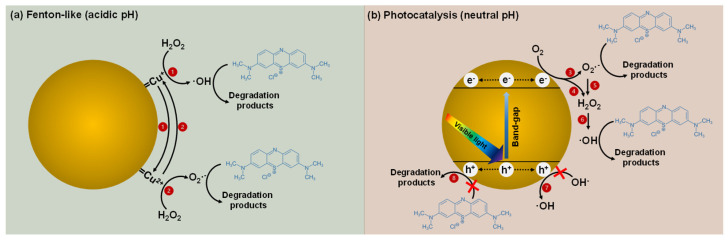
Schematic illustration of Fenton-like reaction at pH = 3.0 and photocatalysis at pH = 7.0 under visible light to generate reactive species and MB degradation process. Numbers in red circles are possibles pathway reactions.

**Table 1 nanomaterials-14-01087-t001:** Main FTIR bands in pristine fabric and F-CuNP.

Wavenumber (cm^−1^)			
Pristine Fabric	F-CuNP	Vibration	Reference
3339	-	Intra-molecular hydrogen bonding C(3)OH⋯O(5) or C(6)O⋯(O)H	[40]
3298	-	Inter-molecular hydrogen bonding C(3)OH⋯C(6)O	[40]
2899	-	Asymmetric stretching vibration of —CH_2_	[41]
2852	-	Symmetric stretching vibration of —CH_2_	[41]
-	1717	C=O stretching	[42]
1638	-	O–H bending of adsorbed water	[40]
-	1506	C=C–C aromatic stretching	[42]
-	1472	-CH_2_ bending of methylene chains in lipids	[43]
1458	1458	O–H in-plane deformation	[40]
1430	-	CH_2_ scissoring	[44]
-	1409	C–OH stretching of alcohol	[45]
1368	-	C–OH bending	[42]
1338	1339	C–O stretching	[42]
1317	-	C–N stretch of aromatic amines	[46]
1205	-	C–O stretching	[40]
-	1176	Phenolic alcohol OH stretching	[45]
1162	-	Anti-symmetrical bridge C–O–C stretching	[44]
-	1117	–CN stretching	[47]
1108	1102	C–OH bending	[45]
1053	-	Stretching vibration of C–O–C in the pyranose skeletal ring	[15]
1026	1022	C–OH groups of cellulose	[40]
1003	-	C–O–H stretching vibration	[48]
986	972	C–O and ring stretching modes	[40]
900	901	Glycosidic deformation –C_1_–O–C_4_ characteristic of the β-glycosidic bond of cellulose	[44]
-	874	C–H out of plane glucose ring	[49]
-	850	Aromatic C-H out-of-plane deformation	[42]
-	793	C–H bending	[50]

## Data Availability

Data have been included in the present paper.

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
