# Peer review of "In Situ Synthesis of Cu2O Nanoparticles Using Eucalyptus globulus Extract to Remove a Dye via Advanced Oxidation"

_nanomaterials, 2024, doi:10.3390/nano14131087_

Round 1
Reviewer 1 Report
Comments and Suggestions for Authors
In this manuscript, Salgado and co-workers investigated the photocatalytic activity of Cu2O on cellulose-based fabric in the degradation of methylene blue (MB). Overall, the manuscript is well-organized, and onle minor revisions are required:
1. I would like to understand the value of specific surface area for the prepared nanoparticles.
2. I do not see much difference in effeciency between photocatalytic and Fenton-like strategies. What is the point of adding hydrogen peroxide?
3. The authors should explain the poor stability of the material and suggest strategies to improve it.
4. The authors focused on the degradation of MB. However, more promising and interesting is the study of the decomposition of other emerging pollutants, such as phenols and antibiotics. I would recommend to provide preliminary data on the degradation of other pollutants.
Comments on the Quality of English LanguageMinor editing is required.
Author Response
Dear Editor,
Attached you can find the answer for Reviewer 1.
With my best regards,
Gladys Vidal

Reviewer 2 Report
Comments and Suggestions for Authors
In this manuscript, the authors reported the synthesis of Cu2O on cellulose-based fabric using Eucalyptus globulus leaf extracts and its application for degradation of methylene blue through both photocatalysis and heterogeneous Fenton-like reactions. These results not only advance our understanding of catalytic processes but also hold promise for sustainable water treatment solutions, which could contribute to environmental protection and resource conservation. Overall, this work has good novelty, and the results were thorough and well organized. This work is therefore considered to be suitable for the journal Nanomaterials. However, in order to further improve the manuscript, some certain revision is required. Please properly address the comments detailed below.
1. The title is a bit confusing. “In situ synthesis of Cu2O nanoparticles using Eucalyptus globulus extract on fabric cellulose-based to removal …”. Is there anything missing after the word “cellulose-based” since it appears to be an adjective? The use of “to removal” is not grammatically sound.
2. When background information is given for advanced oxidation processes, related works can be included in the Introduction (e.g., ACS Sustain Chem Eng, 2022, DOI: 10.1021/acssuschemeng.1c07605; Trends in Chemistry, 2019, DOI: 10.1016/j.trechm.2019.05.006).
3. In line 292, the authors discussed that “By Debye–Scherrer equation the average crystalline size of F-CuNP was determined from the diffractogram being 97.21 nm.” Based on the Debye–Scherrer equation (Equation 3), a certain diffraction angle is required to do the calculation. Which diffraction angle did the authors apply to get the result of “97.21 nm” and why chose this diffraction angle?
4. In Experimental section, TGA was described to have been conducted in a nitrogen atmosphere. Then, is it still possible for Cu2O to be oxidized between 250-350 °C, as claimed by the authors (line 362)?
5. Figure 8, error bars were given for the data presented here. What do these error bars refer to? Please include this information in the Figure caption. The sample applies for Figures 11-12.
Comments on the Quality of English LanguagePlease carefully check the writing throughout the manuscript.
1. Line 24-26, there are two verbs in the sentence “Cu2O nanoparticles on fabric cellulose-based was characterized by SEM-EDS, XRD, Raman, FTIR, UV–Vis DRS, and TGA elucidated Cu2O morphology, structure, and band-gap in this nanoparticles.”
2. Check other writing mistakes carefully, for instance, line 304 “These peaks confirming” and line 305 “These analyzes”.
Author Response
Dear Editor,
Attached you can find the answer for Reviewer 2.
With my best regards,
Gladys Vidal
